# Influence of Annealing on Mechanical Behavior of Alumina-Tantala Nanolaminates

**DOI:** 10.3390/ma16083207

**Published:** 2023-04-19

**Authors:** Helle-Mai Piirsoo, Taivo Jõgiaas, Kaupo Kukli, Aile Tamm

**Affiliations:** Institute of Physics, University of Tartu, W. Ostwaldi 1, 50411 Tartu, Estonia

**Keywords:** atomic layer deposition, nanoindentation, hardness, elastic modulus

## Abstract

Mechanical properties of thin films are significant for the applicability of nanodevices. Amorphous Al_2_O_3_-Ta_2_O_5_ double and triple layers were atomic layer-deposited to the thickness of 70 nm with constituent single-layer thicknesses varying from 40 to 23 nm. The sequence of layers was alternated and rapid thermal annealing (700 and 800 °C) was implemented on all deposited nanolaminates. Annealing caused changes in the microstructure of laminates dependent on their layered structure. Various shapes of crystalline grains of orthorhombic Ta_2_O_5_ were formed. Annealing at 800 °C resulted in hardening up to 16 GPa (~11 GPa prior to annealing) in double-layered laminate with top Ta_2_O_5_ and bottom Al_2_O_3_ layers, while the hardness of all other laminates remained below 15 GPa. The elastic modulus of annealed laminates depended on the sequence of layers and reached up to 169 GPa. The layered structure of the laminate had a significant influence on the mechanical behavior after annealing treatments.

## 1. Introduction

Thin films are important components of nanodevices such as microelectromechanical systems (MEMSs) [1,2]. For MEMS devices, the mechanical properties of the components are significant, as durability and operations are directly influenced [2]. The fine structure of a material influences its mechanical properties, allowing one to manipulate its performance by modifying its composition and structure [3,4]. Deposition process-mediated nanoengineering could provide materials with high mechanical strength [3], hardness [5], flexibility [6,7], or ductility [8] suitable for MEMS applications. 

Atomic layer deposition (ALD) enables, with high accuracy, the control of the layered structure of nanolaminates and, therefore, the physical properties of the films [3]. For example, a layered structure has allowed variations in the hardness of ALD-grown Al_2_O_3_/ZnO nanolaminates from 7 to 11 GPa, surpassing the hardness of single Al_2_O_3_ and ZnO films [9]. The mechanical properties of ALD-grown Al_2_O_3_/ZrO_2_ nanolaminates depended on the nanostructure [10,11]. The layer thickness in ALD-grown Al_2_O_3_/TiO_2_ influenced residual stress in the film in addition to other mechanical properties [12]. 

Post-deposition annealing of thin films can be applied to alter their microstructure by causing crystallization [13,14], diffusion [15], and chemical reactions [16]. For example, the annealing of thin Al_2_O_3_ film has led to crystallization in the temperature range of 800–1200 °C [14,17]. Different phase compositions were obtained, and notably, mechanical stresses introduced during the deposition process [18] could be modified with annealing [17,19]. Hexagonal, orthorhombic, and monoclinic phases of Ta_2_O_5_ have been obtained after annealing amorphous Ta_2_O_5_ films at temperatures higher than 600 °C [20,21,22,23]. For crystalline Ta_2_O_5_, the hardness of 14 GPa has been measured, which is 6 GPa higher than that of the amorphous material [24,25]. However, annealing treatments can also cause a decline in hardness [25,26]. 

Dimensional parameters of nanolaminates influence processes taking place during annealing. For example, in ZrO_2_/SiO_2_ nanolaminates, decreasing constituent layer thicknesses have increased the crystallization temperature of zirconia [27]. Similarly, in TiO_2_/Al_2_O_3_ laminates, the thickness of titania layers influenced its crystallization temperature, and the level of internal tensile stress recovery was also affected by the periodicity of the laminate [12,28]. In HfO_2_/La_2_O_3_ laminates, detectable crystallization occurred only in the topmost hafnia layer at 800 °C, yet an elemental diffusion took place at lower interfaces in the film bulk, as well [15]. In HfO_2_/Y_2_O_3_ laminates, the crystallographic growth orientations of cubic hafnia and yttria, upon annealing, depended on the layer thickness [29,30]. In ZnO/Al_2_O_3_ nanolaminates, the preferred crystallographic orientation was developed in otherwise polycrystalline ZnO grown between amorphous Al_2_O_3_ layers after annealing for 10 min at 800 °C, as solid-state reactions took place at interfaces [16].

To date, studies on mechanic properties of multilayers consisting of amorphous and nanocrystalline materials have directly been devoted to alternately deposited metal–metal alloy [31,32,33] or, more rarely, to metallic-glassy ceramic [34] thin films. It has been shown that the hardness as well as elastic modulus can be considerably affected by the composition, ordering, thickness period, and also annealing of the laminated structures of metal-based multilayers. One could observe, for instance, that peak hardness could be enhanced in laminates with sufficiently low single-layer thicknesses. Although one might expect hardening of a material in the case of well-developed materials, alternate layering of presumably softer, disordered, amorphous, and polycrystalline films can result in structures less prone to sliding and fatigue. This is due to the disruption of dislocations characteristic of crystalline films by the constraint of the amorphous intermediate layers. Therefore, it becomes interesting and relevant to investigate the mechanical performance of alternately layered amorphous and crystalline ceramic compounds, since the ceramic may generally be considered as materials of lower plasticity compared to metals. It would be important, however, that the materials constituting the multilayers were initially deposited in the form of amorphous films due to their higher smoothness and more uniform thickness, compared to those of the polycrystalline ones. One component of such laminated films, i.e., the one with lower crystallization temperature, should then become crystallized during post-deposition heat treatment, while the other component remains amorphous. Earlier indentation studies devoted to ALD-grown oxide multilayers consisting of ZrO_2_-Al_2_O_3_ [10,11], ZrO_2_-HfO_2_ [35], ZrO_2_-SnO_2_ [36], TiO_2_-Al_2_O_3_ [12], Ta_2_O_5_-HfO_2_ [37], Ta_2_O_5_-ZrO_2_ [37], and Ta_2_O_5_-Al_2_O_3_ [37,38] have been carried out on as-deposited films without post-growth engineering of their internal structure. 

As referred to above, Ta_2_O_5_ films can be crystallized at markedly lower temperatures compared to those of Al_2_O_3_. In the present study, post-deposition rapid thermal annealing was conducted on amorphous Ta_2_O_5_-Al_2_O_3_ laminates at temperatures sufficiently high for the crystallization in initially amorphous Ta_2_O_5_ films. The component films were grown by ALD using well-established processes based on trimethylaluminum and tantalum pentaethoxide as metal precursors, ensuring amorphicity as well as appreciable large-area uniformity in terms of composition and thickness. In our previous study, it was shown that the sequence of constituting layers of a nanolaminate affected its mechanical hardness [38]. The results led to a proposal that annealing processes could provide useful changes to the deformation processes and interaction of the layers under mechanical loading. The effect of the alternate layering of amorphous Al_2_O_3_ and crystalline Ta_2_O_5_ on the mechanical behavior of nanolaminates was studied thereafter. The study will add knowledge about the behavior of thin ceramic coatings of artificially engineered composition.

## 2. Materials and Methods

Process parameters set for the ALD of Al_2_O_3_ and Ta_2_O_5_ are summarized in Table 1 and additional information about the deposition of laminates can be found in our previous study [38]. The layered structure of the four laminates, as deposited, are schematically represented in Figure 1. Reference Al_2_O_3_ and Ta_2_O_5_ with thicknesses of 69 nm and 61 nm, respectively, were deposited as well. Rapid thermal annealing was carried out with an in-house-built cylindrical oven with open ends in an air environment for 10 min at 700 and 800 °C.

Oxygen-to-metal atomic ratios in the deposited films were determined with Rigaku wavelength dispersive X-ray fluorescence spectroscope (XRF) ZSX-400. The layered structures of the laminates were dimensionally characterized by X-ray reflectivity (XRR) method using the Rigaku SmartLab^TM^ diffractometer. The surface morphology of the films was characterized using FEI Helios NanoLab 600 scanning electron microscope (SEM). The microscope was equipped with a focused ion beam (FIB), enabling us to also study the cross-sections of the films. A protective platinum coating was deposited onto the sample in the microscope prior to FIB cutting. The surface of the cross-section was polished with an ion beam at 30 kV and 0.28 nA and then viewed with the electron beam in high-resolution mode. Energy dispersive X-ray spectroscopy (EDX) was performed with an Oxford Instruments INCA Energy 350 detector in the Helios NanoLab 600 microscope. Grazing incidence X-ray diffraction (GIXRD) was performed with the Rigaku SmarLab^TM^ diffractometer at 0.6° grazing angle with 0.04° step at 3°/min. 

The Bruker Hysitron TriboIndenter TI980 with a Berkovich tip (triangular pyramid) was used for nanoindentation measurements to determine the hardness and complex elastic modulus of the films. The calibration of the TriboIndenter was carried out prior and following the tests on samples on fused quartz (Figure 2). TriboScan software calculated the mechanical properties based on the tip calibration results and nanoindentation theory developed by Oliver and Pharr [39]. A total of 30 indentations with a maximum load of 0.5 mN in continuous stiffness mode were carried out on each sample. Indentation marks and surface roughness were characterized with the TriboIndenter TI980 with the scanning probe microscopy (SPM) method implementing the same Berkovich tip.

## 3. Results

### 3.1. Microtructure and Elemental Composition

Figure 3 depicts the cross-sections of the Ta_2_O_5_/Al_2_O_3_/Si sample before and after annealing at 800 °C. The layered structure can be recognized after visual distinction between constituent layers on the basis of slight changes in image contrast at the interfaces of the two oxides. Apparent layer thicknesses corresponded with the ones found by XRR (Figure 1). No changes to the layered structure, in terms of thicknesses, were detected after annealing. 

As-deposited films exhibited a homogeneous featureless surface, as expected in the case of amorphous materials (Figure 4). RMS roughness determined with SPM (Berkovich tip) was 2 Å for all as-deposited films, while XRR roughness varied from 7 to 10 Å in the case of laminates and was 4 and 8 Å for Al_2_O_3_/Si and Ta_2_O_5_/Si, respectively.

Annealing caused changes to the surface morphology of the films as recorded with SEM (Figure 4, Figure 5, Figure 6 and Appendix A); however, RMS roughness recorded with SPM only increased slightly after annealing at 700 °C (maximum of 5 Å on Ta_2_O_5_/Al_2_O_3_/Ta_2_O_5_/Si), while annealing at 800 °C resulted in RMS surface roughness varying from 6 to 9 Å for all the films. Variance in SEM contrast in Figure 4, Figure 5 and Appendix A is probably due to the different phase composition of partially crystallized Ta_2_O_5_, as EDX showed uniform spatial distribution of elements (Figure 5). In addition, XRF found the oxygen-to-metal atomic ratios to correspond to those in stoichiometric Al_2_O_3_ and Ta_2_O_5_ for the as-deposited [38] and annealed films. 

The surface morphology of Ta_2_O_5_/Si annealed at 700 °C possessed irregular grain-like features (Figure 4a), whereas Ta_2_O_5_/Al_2_O_3_/Ta_2_O_5_ laminate seemed to contain more oblong-shaped grains, grown probably deeper in the film bulk, in addition to the irregularly shaped grains formed and visible in the surface layers (Figure 4b). The Al_2_O_3_/Ta_2_O_5_/Al_2_O_3_ film contained rectangular, less densely formed grains with a different surface topography, probably as the visible Ta_2_O_5_ grains are covered with an amorphous Al_2_O_3_ layer (Appendix A). Grain-like features possessing similar surface texture as Ta_2_O_5_/Si and Ta_2_O_5_/Al_2_O_3_/Ta_2_O_5_/Si were sometimes visible on the Ta_2_O_5_/Al_2_O_3_/Si sample annealed at 700 °C (Appendix A). Reference Al_2_O_3_/Si and Al_2_O_3_/Ta_2_O_5_/Si exhibited surfaces similar to the as-deposited amorphous films after heating at 700 °C, as is expected, since the 40 nm Al_2_O_3_ layer remaining amorphous at given temperatures could cover the changes in the lower Ta_2_O_5_ layer from SEM viewing in the case of Al_2_O_3_/Ta_2_O_5_/Si. 

The surface morphology of the Ta_2_O_5_/Si, Ta_2_O_5_/Al_2_O_3_/Ta_2_O_5_/Si, and Ta_2_O_5_/Al_2_O_3_/Si appeared more uniform after annealing at 800 °C, with examples presented in Figure 6. The uniform surface with the absence of grain-like features is probably due to the complete crystallization of the Ta_2_O_5_ top layers. Cracks were visible on the surface of these laminates, while the amount and size of the cracks seemed to be larger in the reference Ta_2_O_5_/Si. Surfaces characterized with scanning probe microscopy (SPM) correlated with SEM results; Figure 6 and Figure 7 show similar development of cracks. Cracks also appear more defined on Ta_2_O_5_/Si with SPM compared to laminates with the topmost Ta_2_O_5_ layer (Figure 7). 

The surfaces of Al_2_O_3_/Si and Al_2_O_3_/Ta_2_O_5_/Si remained characteristic of the as-deposited amorphous surface, without distinctive grain-like features apparent after annealing at 800 °C, while the Al_2_O_3_/Ta_2_O_5_/Al_2_O_3_ laminate showed irregular grain-like features unlike on other samples (Appendix A). Possibly, the crystallization of the intermediate Ta_2_O_5_ layer affected the surface of the 27 nm thick Al_2_O_3_ layer, but not the 40 nm thick Al_2_O_3_ layer in the case of Al_2_O_3_/Ta_2_O_5_/Al_2_O_3_/Si and Al_2_O_3_/Ta_2_O_5_/Si, respectively. Films based on Al_2_O_3_-Si interfaces suffered from blistering after annealing. 

### 3.2. Phase Composition

Diffractograms in Figure 8 and Figure 9 describe the changes in the structure of the laminates due to annealing. Processing at 700 °C left Al_2_O_3_/Si and double-layered laminates X-ray-amorphous, while the orthorhombic Ta_2_O_5_ (PDF 00-025-0922) structure was formed in the reference Ta_2_O_5_ film as well as in the triple-layered laminate Ta_2_O_5_/Al_2_O_3_/Ta_2_O_5_. The triple-layered film containing a single Ta_2_O_5_ layer possessed a structure with distinguishable 1 11 0 and 1 11 1 reflections. These findings are supported by SEM studies, as the triple-layered films exhibited larger and more densely populated grain-like features compared to the double-layered films (Figure 4b and Appendix A). Diffractograms of hexagonally structured Ta_2_O_5_ (PDF 00-018-1304) can exhibit reflections overlapping with those of the orthorhombic Ta_2_O_5_ and the formation of hexagonal Ta_2_O_5_ should thus not be excluded. However, earlier in situ TEM studies on Ta_2_O_5_ crystallization with similar grain shapes (Figure 4b) have confirmed the presence of the orthorhombic structure [22]. 

Thicknesses of layers constituting nanolaminated structures have earlier been found to affect the crystallization temperature. For instance, the crystallization temperature of ZrO_2_ has increased by decreasing its thickness in ZrO_2_/SiO_2_ nanolaminates [27]. The amount of crystallization nuclea is quite likely correlated with structure of a laminate, as it is believed that crystallization centers are located at interfaces [27]. In the present study, heat treatment at 700 °C caused crystallization detectable with GIXRD in the triple-layered laminates, yet not in the double-layered ones (Figure 4b, Figure 8 and Appendix A). The significance of the regions in the vicinity of interfaces is, naturally, doubled for triple-layered laminates as compared to the double-layered ones, even as the layer thickness decreases. 

After annealing at 800 °C, Ta_2_O_5_/Si and all the laminates exhibited reflections characteristic of the orthorhombic structure (Figure 9). For the reference Al_2_O_3_ film, a single reflection apparent at 68° could be attributed to the (440) plane of γ-Al_2_O_3_ (PDF 00-029-0063). The Al_2_O_3_/Ta_2_O_5_/Si sample exhibits a possible 440 reflection (Figure 9b), while the other laminates contain predominantly amorphous Al_2_O_3_. Similarly, in a previous study, in the ALD Al_2_O_3_/ZnO laminates, the 20 nm thick alumina layers remained amorphous after annealing at 800 °C for 10 min in air [16]. 

In the diffractograms, the relative intensities of 001 and 1 11 1 reflections varied, with values shown in Table 2. For Ta_2_O_5_/Si, the relative intensities of the reflections remained unchanged for both temperatures and the double-layered laminates annealed at 800 °C possessed shapes of diffractograms similar to that of Ta_2_O_5_/Si. Relative intensities of the reflections for Ta_2_O_5_/Al_2_O_3_/Ta_2_O_5_/Si evened up with the increase in temperature as the 001 reflection weakened. The 001 signal remained low for the Al_2_O_3_/Ta_2_O_5_/Al_2_O_3_/Si for both annealing processes.

Various shapes for Ta_2_O_5_ grains during the initial stages of crystallization have been observed previously, and the oblong shape (Figure 4b) of grains has been reported to be characteristic of annealed orthorhombic tantala thin film [22]. The elongated shape of the grains was proposed to be the result of a growth along a preferable crystallographic direction [22]. In the present study, the Ta_2_O_5_/Al_2_O_3_/Ta_2_O_5_ laminate contained oblong grains and revealed a relatively strong 001 reflection in the diffractogram compared to those recorded from the rest of the samples with other grain shapes (Figure 4, Figure 8 and Appendix A, Table 2).

From the results, it can be concluded that the crystallization process characterized by crystallization temperature, grain shapes, and relative intensities of reflections was influenced by the amount and order of constituent layers in the Al_2_O_3_-Ta_2_O_5_ laminates. At the same time, the crystallization of Ta_2_O_5_ does not significantly raise the surface roughness of laminates.

### 3.3. Mechanical Properties

Nanoindentation results are shown in graphs, with the Y-axis presenting the tip displacement, i.e., the depth, and the X-axis presenting the mechanical property measured and calculated (Figure 10, Figure 11 and Figure 12). The axis designation differs from the conventional presentation mode, with the mechanical property on the Y-axis and indentation depth on the X-axis, in order to better visualize the changes of the properties when indenting into the depth of the solid films and correlate the results with the layered structure (Figure 11 and Figure 12). Nanoindentation marks are visualized in Figure 7.

The hardness of the as-deposited and annealed (700 and 800 °C) reference oxide films is presented in Figure 10. Crystallization of Ta_2_O_5_ at 800 °C increased the hardness from approximately 8 GPa to 12 GPa, while annealing at 700 °C resulted in a slightly lower average hardness (Figure 10a). The hardness of the crystalline Ta_2_O_5_ material has been reported to reach 14 GPa [24,25]. SEM and SPM studies revealed cracks in the surface of the annealed Ta_2_O_5_/Si (Figure 6a and Figure 7a), which could have caused lower mechanical hardness compared to the literature. Crack formation might have been due to stresses in the film, as annealing can induce residual stresses in Ta_2_O_5_ films [26].

Annealing at both temperatures increased the hardness of Al_2_O_3_/Si from approximately 12 GPa to 18 GPa (Figure 10b). Al_2_O_3_ films ALD-grown at 300 °C have been reported to possess residual tensile stress around 200 GPa, which further decreased upon heating [12,17,18]. In the present study, a possible explanation for the increase in hardness after annealing could be the similar stress reduction in Al_2_O_3_/Si.

The average difference in hardness between the reference Ta_2_O_5_ and Al_2_O_3_ in as-deposited states annealed at 700 and 800 °C was 4, 7, and 6 GPa, respectively. Crystallization and blistering probably caused an increase in the standard deviation of the nanoindentation results. The hardness of the Si substrate was 13.5 ± 0.4 GPa.

One can see in Figure 11a that the heat treatment of Al_2_O_3_/Ta_2_O_5_/Si at 700 °C increased the hardness near the surface slightly compared to the as-deposited hardness, while 800 °C of heat increased the overall hardness markedly, up to 14 GPa, and still slightly hardened the surface. Figure 11b presents the hardness of Ta_2_O_5_/Al_2_O_3_/Si, where annealing at 700 °C left the mechanical hardness of the material unchanged, while annealing at 800 °C increased it to 16 GPa throughout the depth measured, i.e., to a value exceeding that of all laminates. 

Figure 12a presents the hardness of the Al_2_O_3_/Ta_2_O_5_/Al_2_O_3_ laminate. Annealing at 700°C only slightly increased the average hardness near the surface, while the deviation from the average value increased compared to the as-deposited material. Treatment at 800 °C increased the hardness up to approximately 13 GPa. The hardness of Ta_2_O_5_/Al_2_O_3_/Ta_2_O_5_/Si (Figure 12b) was similarly weakly affected by the temperature of 700 °C, while the higher temperature caused an increase to around 13 GPa.

Elastic moduli varied weakly along the depth of the films. Moduli at 10 nm from the surfaces of Al_2_O_3_ and Ta_2_O_5_ films deposited on silicon were ~155 and ~140 GPa (Figure 13), respectively. Annealing at 800 °C raised the modulus of Al_2_O_3_/Si to ~170 GPa near the surface, while already at 15 nm into the film depth, it became comparable to that of the as-deposited material. The Si substrate possessed a modulus of 147 ± 3 GPa. Annealing caused the modulus of the Ta_2_O_5_ film to reach 155 GPa. 

As-deposited laminates possessed an elastic modulus ~150 GPa along the whole depth of the film (Figure 13). Annealing at 700 °C increased the moduli of Ta_2_O_5_/Al_2_O_3_/Si and Al_2_O_3_/Ta_2_O_5_/Al_2_O_3_/Si to approximately 155 GPa and left the moduli unchanged for the other laminates. Annealing at 800 °C increased the moduli of Al_2_O_3_/Ta_2_O_5_/Si to 170 GPa, comparable to the reference Al_2_O_3_ film on silicon. Other laminates, after the same thermal treatment, possessed slightly lower elastic moduli (Figure 13). 

## 4. Discussion

### 4.1. Mechanical Behavior of Laminates after Annealing at 700 °C for 10 min

After annealing at 700 °C, the near-surface hardness of amorphous laminates with a top Al_2_O_3_ layer increased (Figure 11 and Figure 12). The hardness was not affected by the bottom Al_2_O_3_ layers in annealed laminates compared to that of the as-deposited material (Figure 11 and Figure 12). At the same time, the elastic modulus increased after 700 °C for laminates with bottom Al_2_O_3_ layers (Figure 13). The increase in hardness for the top Al_2_O_3_ layers in laminates and, similarly, for that in the reference Al_2_O_3_ film on Si might have been caused by the relaxation of residual tensile stresses due to the internal diffusion at elevated temperatures [12,16,17,18]. However, for the bottom Al_2_O_3_ layers, the stress relaxation could be suppressed by interfaces with both the substrate and the upper Ta_2_O_5_ film. The overall residual stress in thin films is affected by laminated structure [12,28] and residual stresses can influence the hardness and elastic modulus of thin films [40]. After annealing, various stress levels might be present in constituting layers in the laminates in the present study and, thus, affect the hardness and modulus differently, causing their variance. 

Significant crystallization after annealing at 700 °C occurred in Ta_2_O_5_/Al_2_O_3_/Ta_2_O_5_ laminate and, as expected, the average overall hardness increased compared to that of the as-deposited laminate (Figure 12b). The hardness–depth curve of the annealed laminate remained similar to that of the as-deposited laminate, indicating similar deformation and stress distribution between neighboring layers during loading. Probably, the intermediate Al_2_O_3_ layer was mechanically harder compared to the surrounding, partially crystallized Ta_2_O_5_ layers. Additionally, the modulus of Ta_2_O_5_/Al_2_O_3_/Ta_2_O_5_/Si was unaffected by the applied treatment temperature of 700 °C, unlike that of the reference Ta_2_O_5_/Si. The lower modulus of the laminate could be caused either by the influence of intermediate Al_2_O_3_ or the preferred orientation of Ta_2_O_5_ crystallites (Figure 4 and Figure 8, Table 2), which would require further detailed studies before the conclusive specifications. 

The results indicate that the sequence of layers, layer thickness, occurrence of crystallization, and crystal growth orientation affect the mechanical properties of a laminate after annealing.

### 4.2. Mechanical Behavior of Laminates after Annealing at 800 °C for 10 min

After annealing at 800 °C, the highest average hardness was measured for Ta_2_O_5_/Al_2_O_3_/Si (Figure 11b). The hardness of crystalline Ta_2_O_5_, 14 GPa [25], is ~2 GPa higher than the hardness of the reference Ta_2_O_5_ film on Si (Figure 10a). The hardness of Ta_2_O_5_/Si might have been lowered by cracks in the films. Figure 6a and Figure 7a depict cracks observable in Ta_2_O_5_/Si, which could possibly be caused by tensile stresses induced during annealing. It has been previously demonstrated that layering crystalline material alternately with amorphous Al_2_O_3_ can reduce residual tensile stresses in thin films [12]. Layering in the Ta_2_O_5_/Al_2_O_3_/Si and Ta_2_O_5_/Al_2_O_3_/Ta_2_O_5_/Si samples could have reduced the tensile stresses and led to fewer and less distinct cracks compared to those in the reference Ta_2_O_5_/Si (Figure 6 and Figure 7). The latter, in turn, might have caused the increase in hardness of Ta_2_O_5_/Al_2_O_3_/Si. The higher hardness of Ta_2_O_5_ in the Ta_2_O_5_/Al_2_O_3_/Ta_2_O_5_ laminate on Si may have leveled the hardness along the depth compared to the wavelike curve of the hardness–depth function of the as-deposited film (Figure 12b). 

The second double-layered Al_2_O_3_/Ta_2_O_5_ laminate possessed the highest modulus, which was comparable to that of the reference Al_2_O_3_/Si (Figure 13). The annealed Al_2_O_3_/Ta_2_O_5_/Si exhibited possible γ-Al_2_O_3_ reflections, similar to the reference film (Figure 9b), which is the most probable cause of the increase in the modulus. The sequence of oxide layers from surface to substrate in the double-layered laminates influenced the hardness and modulus after annealing at 800 °C.

## 5. Conclusions

Atomic layer-deposited amorphous Al_2_O_3_-Ta_2_O_5_ nanolaminates on silicon, grown to an overall thickness of ~70 nm, were subjected to rapid thermal annealing at 700 and 800 °C in air for 10 min. The nanolaminates consisted of either two or three layers, while the sequence of constituent Al_2_O_3_ and Ta_2_O_5_ layers was alternated (Figure 1). 

The multilayered structure influenced the crystallization processes in Ta_2_O_5_ layers, as the grain shape, relative intensities of reflections of diffractograms, and the crystallization temperature were affected by the layer thickness and sequence. Annealing at 700 °C resulted in partially crystallized Ta_2_O_5_ in triple-layered laminates, as SEM images revealed areas with different contrast; however, EDX showed a homogeneous distribution of Al, Ta, and O. XRD confirmed the crystallization of orthorhombic Ta_2_O_5_ in triple layers, while double layers remained amorphous. Ta_2_O_5_ crystallized more uniformly after treatment at 800 °C and surfaces remained relatively smooth, less than 10 Å.

Annealing at both temperatures increased the hardness of the reference Al_2_O_3_ film to ~18 GPa and the Ta_2_O_5_ film to ~12 GPa. The sequence of layers in laminates influenced their near-surface hardness and elastic modulus after annealing at 700 °C. The Ta_2_O_5_/Al_2_O_3_ laminate possessed the highest hardness values, ~16 GPa, after annealing at 800 °C, whereas the average hardness of other laminates remained below 15 GPa. The hardness of crystalline Ta_2_O_5_ in the Ta_2_O_5_/Al_2_O_3_ laminate was most likely increased by the stabilizing influence from underlaying amorphous Al_2_O_3_, which reduced crack development in Ta_2_O_5_. The Al_2_O_3_/Ta_2_O_5_ laminate possessed the highest elastic modulus, 169 GPa, after annealing at 800 °C. Both the thickness and the sequence of layers affected the processes taking place during annealing, influencing the mechanical behavior of the laminates. 

## Figures and Tables

**Figure 1 materials-16-03207-f001:**
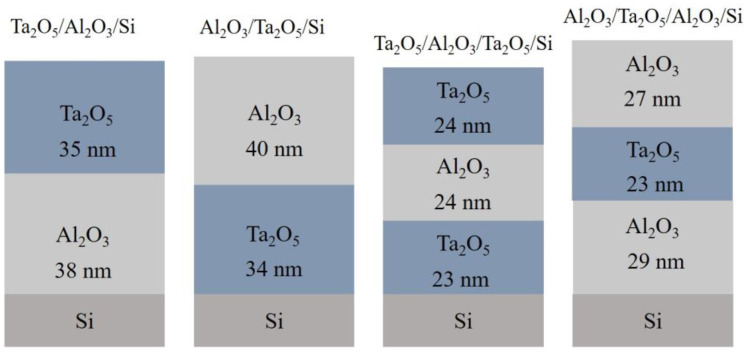
Schematics of the layered structure of the investigated laminates with notations shown above. Layer thickness determined with XRR with error ± 1 nm.

**Figure 2 materials-16-03207-f002:**
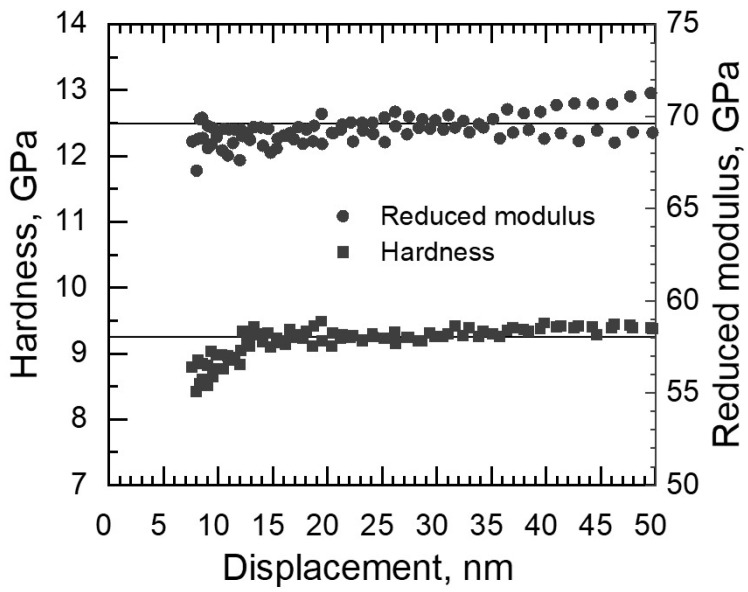
Nanoindentation test measurement results on fused quartz with hardness of 9.25 GPa and reduced modulus of 69.6 GPa (horizontal lines).

**Figure 3 materials-16-03207-f003:**
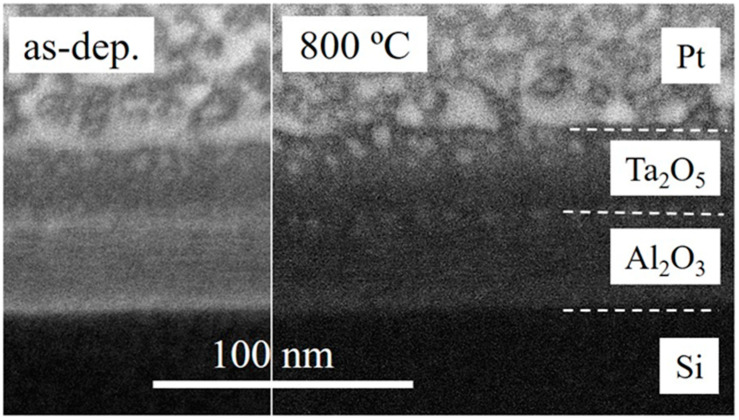
SEM images of FIB-made cross-sections of as-deposited and annealed Ta_2_O_5_/Al_2_O_3_/Si sample with a protective platinum coating.

**Figure 4 materials-16-03207-f004:**
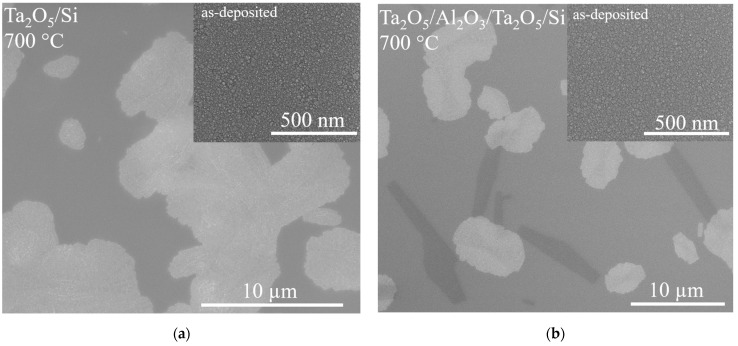
SEM images of the surface of laminates as deposited and after annealing at 700 °C: (**a**) Ta_2_O_5_/Si; (**b**) Ta_2_O_5_/Al_2_O_3_/Ta_2_O_5_/Si.

**Figure 5 materials-16-03207-f005:**
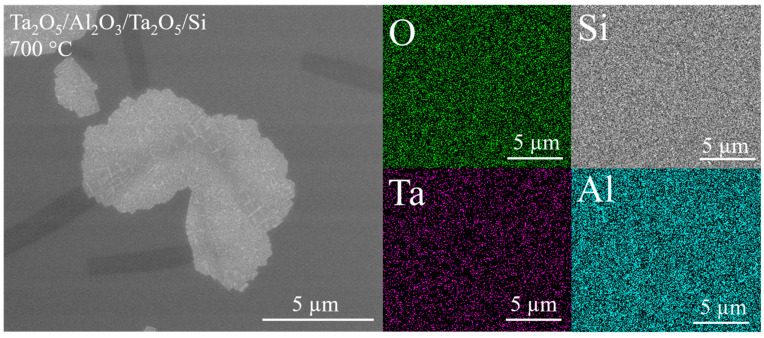
Spatial element (O, Si, Ta, Al) distribution measured with EDX on grain-like surface features of Ta_2_O_5_/Al_2_O_3_/Ta_2_O_5_ film after annealing at 700 °C.

**Figure 6 materials-16-03207-f006:**
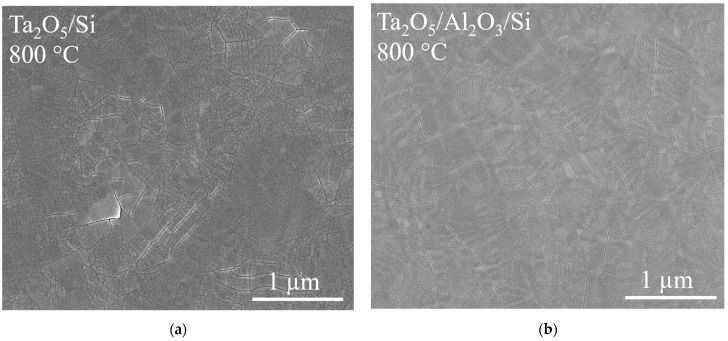
SEM images of the surface of laminates after annealing at 800 °C: (**a**) Ta_2_O_5_/Si; (**b**) Ta_2_O_5_/Al_2_O_3_/Si.

**Figure 7 materials-16-03207-f007:**
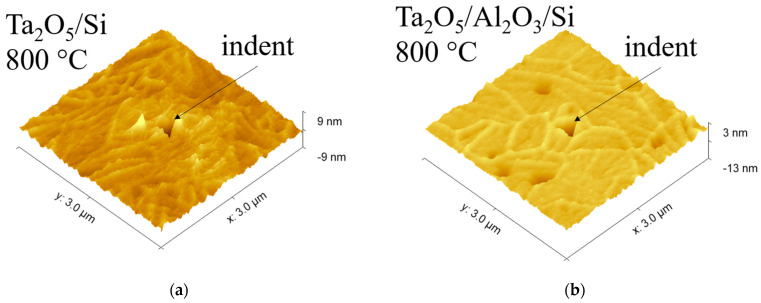
SPM images of the surface of laminates after annealing at 800 °C: (**a**) Ta_2_O_5_/Si; (**b**) Ta_2_O_5_/Al_2_O_3_/Si. Images depict plastic deformation marks left in the material by the triangular pyramid-shaped Berkovich tip during nanoindentation.

**Figure 8 materials-16-03207-f008:**
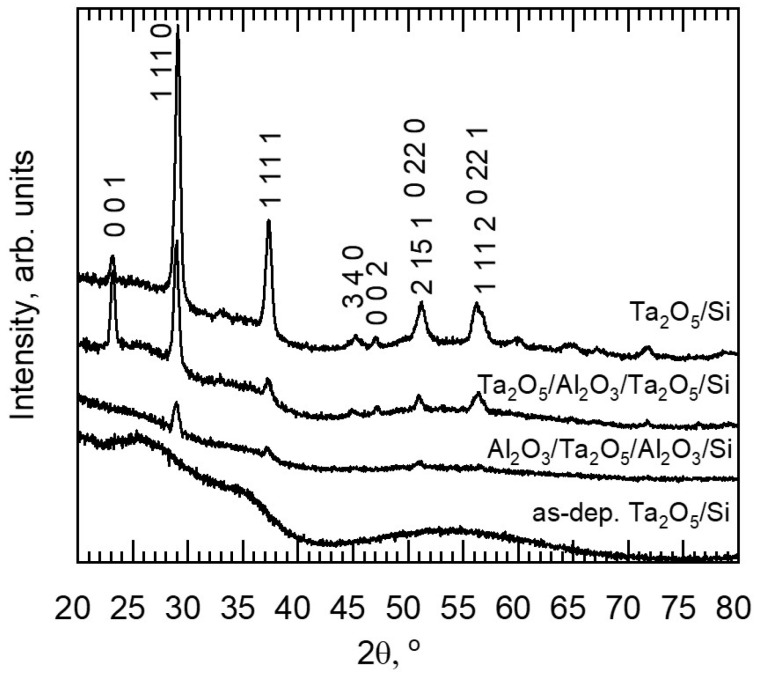
Diffractograms of as-deposited Ta_2_O_5_/Si and three films that revealed crystallinity after annealing at 700 °C. Reflection indices of orthorhombic Ta_2_O_5_ (PDF 00-025-0922) noted.

**Figure 9 materials-16-03207-f009:**
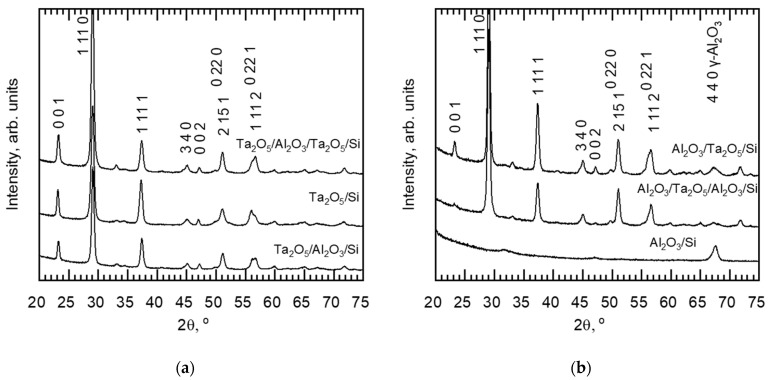
Diffractograms of all the films annealed at 800 °C: (**a**) Ta_2_O_5_/Al_2_O_3_/Ta_2_O_5_/Si, Ta_2_O_5_/Si, and Ta_2_O_5_/Al_2_O_3_/Si; (**b**) Al_2_O_3_/Ta_2_O_5_/Si, Al_2_O_3_/Ta_2_O_5_/Al_2_O_3_/Si, and Al_2_O_3_/Si. Reflection indices of orthorhombic Ta_2_O_5_ (PDF 00-025-0922) and one of γ-Al_2_O_3_ (PDF 00-029-0063) noted.

**Figure 10 materials-16-03207-f010:**
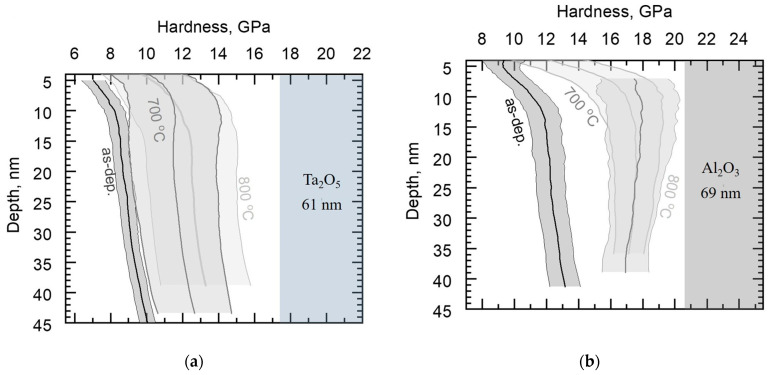
Average hardness along the film depth, with standard deviation of the layered structure of the film schematized on the right side of the panel: (**a**) as-deposited and annealed Ta_2_O_5_/Si; (**b**) as-deposited and annealed Al_2_O_3_/Si.

**Figure 11 materials-16-03207-f011:**
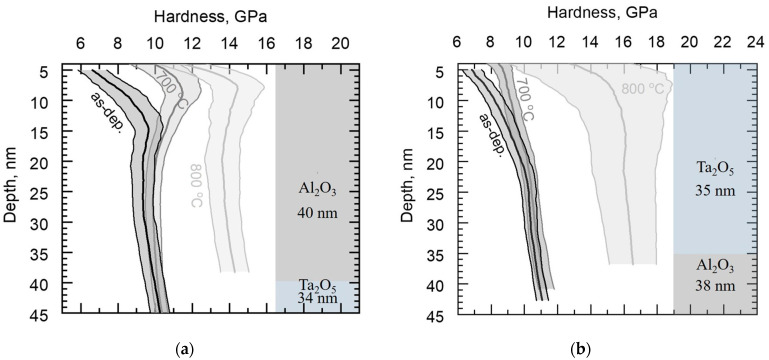
Average hardness along the film depth, with standard deviation of the layered structure of the film schematized on the right side of the panel: (**a**) as-deposited and annealed Al_2_O_3_/Ta_2_O_5_/Si; (**b**) as-deposited and annealed Ta_2_O_5_/Al_2_O_3_/Si.

**Figure 12 materials-16-03207-f012:**
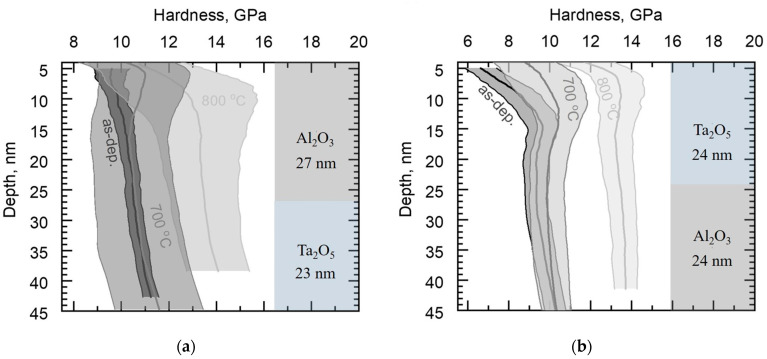
Average hardness along the film depth, with standard deviation of the layered structure of the film schematized on the right side of the panel: (**a**) as-deposited and annealed Al_2_O_3_/Ta_2_O_5_/Al_2_O_3_/Si; (**b**) as-deposited and annealed Ta_2_O_5_/Al_2_O_3_/Ta_2_O_5_/Si.

**Figure 13 materials-16-03207-f013:**
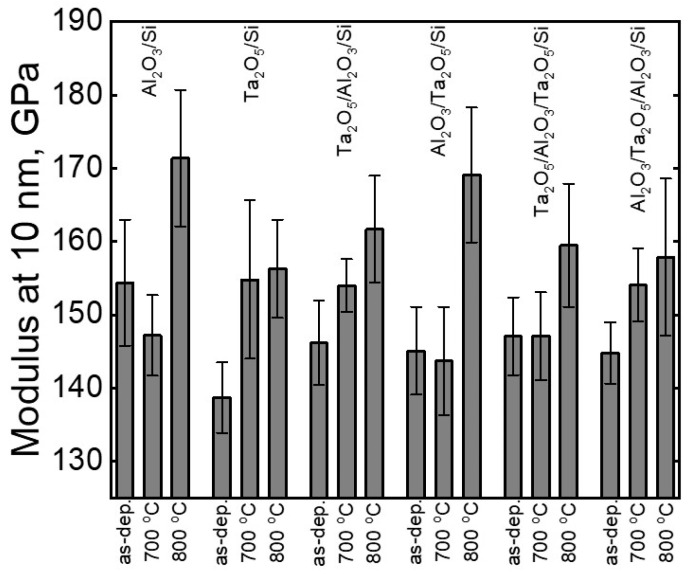
Average elastic moduli with standard deviations at 10 nm for all the measured films in their as-deposited states and after annealing at two different temperatures, as indicated by the notations on X-axis. The stacked structures evaluated by indentation are described by labels above data bars.

**Table 1 materials-16-03207-t001:** Atomic layer deposition parameters for Al_2_O_3_ and Ta_2_O_5_ in a flow-type in-house-built reactor on Si (100) wafer at 300 °C.

Oxide	Metal Precursor Pulse Time	Purge Pulse Time	Oxygen Source Pulse Time	Purge Pulse Time	Growth Rate [38]
Al_2_O_3_	Al(CH_3_)_3_ at 22 ± 1 °C 2 s	N_2_ 2 s	H_2_O at 22 ± 1 °C 2 s	N_2_ 5 s	0.11 nm/cycle
Ta_2_O_5_	Ta(OCH_2_CH_3_)_5_ at 95 ± 5 °C 2 s	N_2_ 2 s	H_2_O at 22 ± 1 °C 2 s	N_2_ 5 s	0.07 nm/cycle

**Table 2 materials-16-03207-t002:** Relative intensities of reflections of diffractograms from Figure 6 and Figure 7.

Temperature	Sample	001	1 11 1
700 °C	Ta_2_O_5_/Si	31	42
	Ta_2_O_5_/Al_2_O_3_/Ta_2_O_5_/Si	84	27
800 °C	Ta_2_O_5_/Si	33	40
	Al_2_O_3_/Ta_2_O_5_/Si	16	37
	Ta_2_O_5_/Al_2_O_3_/Si	31	32
	Al_2_O_3_/Ta_2_O_5_/Al_2_O_3_/Si	5	23
	Ta_2_O_5_/Al_2_O_3_/Ta_2_O_5_/Si	21	18
PDF 00-025-0922		85	75

## Data Availability

The data that support the findings of this study are available from the corresponding author upon reasonable request.

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
