# Peer review of "Influence of Annealing on Mechanical Behavior of Alumina-Tantala Nanolaminates"

_materials, 2023, doi:10.3390/ma16083207_

Round 1

Reviewer 1 Report

The authors prepared Al2O3 -Ta2O5 double- and triple-layers in a previous publication, and they propose the effect of annealing on this layer in this manuscript. There are several points that must be addressed by the authors to evaluate the work.

1.       Firstly, the novelty must be stated clearly.

2.       Revise “to” to “on” in the title.

3.       In the abstract “…………………. constituent layer thickness varying from 40 to 23 nm.” Please Revise this sentence, in which, what is the 40-23 nm layer?

4.       In the introduction, there is not any paragraph about Al2O3 -Ta2O5, such as related applications or previous studies, research gap, aims: why did you choose it, etc.

5.        It is not clear the technique used for Figure 3.

6. In The image of Figure 4, what is the white and dark area, please point out, is not a thin film, and why there are different contrast. “As-deposited films exhibited homogeneous featureless surface, as expected in the case of amorphous materials (Fig. 4)”, please revise, or show the same magnification of Figure 5. Please unify the magnifications of the SEM image, if possible, show a 100 nm scale to understand the roughness well.

7.       If the surface roughness is high, how did you find the error in thickness measurements is +-1 nm by XRR, where the constructive and destructive peaks are sensitive to the surface roughness? (Could you show the XRR spectra of this part).

8.       “Reference Al 2 O 3 /Si and Al 2 O 3 /Ta 2 O 5 /Si did not reveal any grain-like features after heating at 700 °C.” please explain the reason compared to the as-prepared ones.

9.       “  ….. while other laminates consist of amorphous Al2O3.” The author should explain the reason behind this behavior.

10.   The plane directions/Miller indices should be placed in suitable brackets.

11.   In line 192, it is mentioned that “Crystallization of Ta2O5  at 800 °C increased the hardness…..”, then in line 195, it is mentioned that “SEM ….. Cracks…. probably lowered the hardness”. Then 199, annealing increased the hardness. Please revise.

12.   Figures 8-10 are not easy to read since the colors overlap.  

13.   The manuscript lacks equations used for calculations and measured parameters.

14.   In general, the research lacks a deep interpretation of the data, it is only a description of behavior.

15.   As mentioned that the annealing process did not affect the Al2O3 layer, did it affect the hardness?

16.   Figure 6b does not exist. 

Reviewer 2 Report

The manuscript under review is dedicated to formation of single-, double-, and triple-layered Al2O3/Ta2O5 thin films on a Si substrate. As-prepared and annealed films are subjected to comprehensive study in terms of composition, morphology, crystal structure, and mechanical properties. The introduction part provides sufficient background. Experimental details and results are almost clearly presented and discussed. Conclusions supported by the results. Thus, the manuscript under review can be published in Materials after minor revision. The following issues should be commented/corrected: 1. Line 33 – subscript in ZrO2 2. Description of single-layer reference films should be added in Experimental part 3. SEM images, which are described, but not included in section 3.2, can be added in Supporting Information 4. SEM images in Figs 4 and 5 are hard to compare with each other due to different magnification 5. Fig. 4b – what is a reason of dark and bright areas in SEM images: morphology or phase composition? EDX or SEM images in BSE mode may be useful here 6. Line 156 – Fig. 6b is absent in manuscript 7. Figs. 6 and 7 – please, indicate in image or in caption for which phases indexes are belonged to

Reviewer 3 Report

Dear Author, please refer to the following for quality improvement. 

1. Please specify the magnification for all figures. 

2. For morphology, please use FE-SEM rather than SEM since your samples is nanolaminates. 

3. Why Fig. 4(b) and Fig. 5(b) is not the same layers? It should be the same one for comparison. 

4. Not enough discussions for morphology discussion. 

5. Please improve the story board of your manuscript to make it easy to understand. 

Round 2

Reviewer 1 Report

The revised version is sufficient for acceptance.